# Systematic Review of Parotid Gland Sarcomas: Multi-Variate Analysis of Clinicopathologic Findings, Therapeutic Approaches and Oncological Outcomes That Affect Survival Rate

**DOI:** 10.3390/cancers14194862

**Published:** 2022-10-05

**Authors:** Andrea Colizza, Arianna Di Stadio, Massimo Ralli, Pietro De Luca, Carlo Cavaliere, Antonio Gilardi, Federica Zoccali, Mara Riminucci, Antonio Greco, Alessandro Corsi, Marco de Vincentiis

**Affiliations:** 1Department of Sense Organs, Sapienza University of Rome, 00161 Rome, Italy; 2Department GF Ingrassia, University of Catania, Via di Santa Sofia 87, 95123 Catania, Italy; 3Department of Medicine, Surgery and Dentistry, University of Salerno, 84084 Fisciano, Italy; 4Department of Molecular Medicine, Sapienza University of Rome, 00161 Rome, Italy

**Keywords:** parotid gland, sarcoma, head and neck malignancies, surgery, review

## Abstract

**Simple Summary:**

Primary sarcomas of the parotid gland are rare malignancies of mesenchymal origin for which there is no generally well-defined treatment guideline. The aim of this study was to analyze, through the review of the literature, the factors affecting the prognosis of patients with primary sarcoma of the parotid gland. Size/extension at the diagnosis and the sarcoma’s histotype were the most important prognostic factors. Multivariate analysis showed that surgery performed on the tumor was the only parameter affecting long-term survival. In particular, total parotidectomy with preservation or, in the most advanced cases, with the sacrifice of facial nerve should be performed at any age independently of tumor histology.

**Abstract:**

The aim of this study was to systematically review the literature of sarcoma of the parotid gland in order to analyze the main factors affecting survival rate. A systematic literature review was performed between January 1990 to November 2021, and 88 patients affected by parotid gland sarcomas were included. The most common histological types were Rhabdomyosarcoma and Synovial Sarcoma. From our review, it emerges that primary sarcomas of the parotid glands are locally aggressive but show low tendency to metastasize to the lymph nodes of the neck and that surgery (i.e., total or radical parotidectomy) is the main approach for their treatment. The global overall survival (OS) is 52% at 5 years and 34.1% at 10 years. The OS for T1, T2, T3, T4 tumor at 5 years of follow up is 80.0%, 66.5%, 56.7% and 33.3%, respectively. Size/extension at the diagnosis and the sarcoma’s histotype are the most important prognostic factors. Multivariate analysis showed that surgery (total or radical parotidectomy) performed on the tumor (*p* = 0.0008) was the only parameter that significantly affected the OS. Among the other variables, age (younger), use of adjuvant therapy and lymph node metastasis showed borderline significative values (*p* = 0.05). Our analysis suggests that, when a primitive parotid sarcoma is diagnosed, total or radical parotidectomy should be performed at any age independent of tumor histology. Because regional lymph node metastases from parotid sarcomas are uncommon, alternative strategies (e.g., close follow-up by imaging and evaluation of sentinel lymph nodes) should be pursued before lymph node (selective/radical) dissection.

## 1. Introduction

Sarcomas represent an unusually heterogeneous group of tumors originating from mesenchymal cells. Even though they arise most commonly in soft tissues and bone, they may also occur in parenchymatous organs [1,2]. The overall annual incidence of soft tissue sarcomas is less than 1% of all malignant tumors, and about 10% of patients have metastasis at diagnosis, predominantly in the lung [2]. Histological evaluation of the tumor is mandatory for the proper assessment of the diagnosis and, in combination with clinical data, for the staging of the tumor [2,3]. The classification of soft tissue sarcomas includes various types (e.g., adipocytic, fibroblastic and myofibroblastic, vascular, smooth and skeletal muscle, peripheral nerve sheath malignant tumors) and sub-types (e.g., well-differentiated, dedifferentiated, myxoid and pleomorphic liposarcoma; embryonal, alveolar, pleomorphic, and spindle cell rhabdomyosarcoma; spindle cell, biphasic and poorly differentiated synovial sarcoma) [2]. While pathologic diagnosis provides sufficient information for the prediction of the clinical behavior for some sarcomas (e.g., Ewing sarcoma, differentiated and pleomorphic liposarcoma), grading and/ staging are needed for others [2].

In the major salivary glands, primary sarcomas are very rare. Their prevalence has been estimated to be in the range of 0.5 to 1.5% of all salivary gland malignancies and parotid is the most commonly involved salivary gland [4,5,6,7,8]. Although descriptions are essentially limited to case report or small series [4,5], three reviews on this topic have also been published [6,7,8]. Data from these reviews were overall similar and demonstrate adverse outcomes for the affected patients. For example, the metastatic and mortality rates were 38% and 40% in the review of Auclair et al. [6], 40% and 36% in that of Luna et al. [7], and 38% and 28% in that of Cockerill et al. [8].

Because of their rarity, currently there are not well-defined guidelines for treatment of sarcomas arising in the parotid gland [8]. However, because treatments for sarcomas arising at other anatomic sites are expected to be tailored according to the size and extension (invasion of adjacent structures, regional lymph node and distant metastasis) of the tumor and to the specific histological sarcoma type/sub-type and grade, it is likely that these are the same most important predictive factors of oncologic outcome for patients with sarcomas arising in the parotid gland as well [6,9,10,11,12,13].

This study aimed to systematically review the literature reporting primary sarcomas of the parotid glands and report epidemiologic and clinicopathology, therapeutic approaches and outcomes for these rare entities. Compared to the previous published reviews [6,7,8], multivariate analysis was performed to investigate which of the clinicopathologic parameters and therapeutic approaches affect the prognosis of these patients.

## 2. Materials and Methods

### 2.1. Literature Search Strategy

Following the Preferred Reporting Items for Systematic and Meta-Analyses (PRISMA) guidelines, two independent researchers performed a systematic review of the literature using MEDLINE, EMBASE, PubMed, and Scopus databases. The search strategy was conducted using combinations of the following terms: “sarcoma” OR “malignant mesenchymal tumor” OR “soft tissue tumor” OR “mesenchymal tumor” AND “parotid gland”. 

### 2.2. Study Selection

For the scope of this review the international literature produced between January 1990 to November 2021 was screened. The last day of the search was 30 November 2021.

Only studies in English were considered as suitable to be included. Clinical series, case report, observational, longitudinal and clinical trials were considered. To be included, studies had to report data about histological type of tumor, tumor site (superficial lobe, deep lobe or both), TNM according to AJCC [3], treatment strategy, oncological outcome and follow-up for each individual patient. Articles with lacking information regarding the size/extension of tumor, outcome, time of follow up and secondary neoplasms due to previous radiation therapy were excluded.

Title and abstract were watchfully examined by two authors (AN.C and C.C) independently, and disagreements were resolved by a discussion with a third author (MAS.R or A.D.S). We excluded unrelated articles and papers not in the English language. Then the selected articles were analyzed in full. 

### 2.3. Data Extraction

The full text of the included studies was reviewed, and data extraction was performed using a standard registry database. Epidemiologic and clinicopathologic data, registered in each case, included age, gender, clinical features, tumor location in the parotid gland (superficial lobe, deep lobe or both), histopathologic diagnosis, TNM according to AJCC [3], type of surgery performed on the tumor (total or radical parotidectomy), surgery performed on lymph nodes, neoadjuvant or adjuvant treatment, recurrence, metastasis, and survival data. We used the overall survival (OS, time between the first treatment and that of death or last recorded follow-up) as the primary oncological outcome.

### 2.4. Study Quality

The levels of evidence according to the standards by Wasserman et al. [14] of the included articles were scored as follows: Level I: randomized controlled trials; level II: prospective studies with an internal control group; level III: retrospective studies with an internal control group; level IV: case series without an internal control group; and level V: consensus or expert opinion without critical appraisal.

### 2.5. Statistical Methods

Descriptive analyses were mainly applied. Data are indicated as mean, range and percentage. Kaplan–Meier analyses were assessed for outcome analysis. Multilinear regression analyses were performed to identify which of the identified parameters could have an influence on the patients’ Tumor-Free Survival (TFS, time after first treatment during which no sign of tumor is found)) and Tumor-Specific Survival (TSS, time between the first treatment to that of death from tumor). As first, we evaluated the effect of age, gender, presence of positive lymph node, presence of metastasis, tumor histologic type, surgery performed on the gland, surgery performed on lymph node, adjuvant therapy, neoadjuvant therapy and follow-up on the dependent variable TFS; then the same independent variables were analyzed considering TSS as dependent variable. A *p* value < 0.05 was considered as statistically significant. All statistical analyses were performed using SPSS Version 25.0 (IBM Corp, Armonk, NY, USA).

## 3. Results

### 3.1. Search Results, Data Synthesis and Analysis

The search algorithm and review results are outlined in Figure 1. The initial search found 307 studies on the screened library databases. The removal of duplicates identified 302 publications. All of the 302 papers were screened in title and abstract. One hundred and ninety papers were excluded because they were unrelated with this review or written in other languages; 112 manuscripts were reviewed in full text. Of these, 61 studies met the inclusion criteria, while the remaining 51 studies were excluded. The included studies were all published in peer-reviewed journals. No randomized controlled trials were identified. As the included studies were heterogeneous, a formal meta-analysis could not be appropriately performed. The data collected from each study were transcribed in a tabular form.

According to the standards by Wasserman et al. [14], the quality evidence of the studies included in the analysis ranged from 2 to 4 (median 3.25, SD 0.75).

### 3.2. Study Cohort

A total of 88 patients affected by sarcoma of the parotid gland were included. Among them, 48 were males (54.6%) and 40 females (45.4%) with a mean patient age of 39.4 years (range: 1–87 years). The most frequent clinical presentation was a painless, slowly growing swelling in the parotid region (59.1%). Another frequent initial symptom was trismus (19.3%). Other less common symptoms included facial palsy, otalgia, skin fistula and ear fulness. The main demographic characteristics and the initial symptoms of disease for each case are summarized in Table 1.

### 3.3. Sarcoma’s Characteristics

In 49 cases (55.7%), the tumor involved only the superficial lobe of the parotid gland. In seven cases, only the deep lobe (8.0%) was interested. Both superficial and deep lobes, the parapharyngeal space and skull base were interested in 21 (23.9%), 8 (9.1%) and 3 (3.4%) patients, respectively. Nine tumors were staged T1 (10.2%), 28 T2 (31.8%), 24 T3 (27.3%) and 27 T4 (30.7%). Nodal involvement was reported in 7 cases and distant metastasis in 6 (6.8%).

The main characteristics of the tumor are resumed in Table 2.

Nineteen different types of histological diagnosis were observed in our study (Table 3). The most common were Rhabdomyosarcoma (21 cases), Synovial Sarcoma (10 cases) and Angiosarcoma and Leiomyosarcoma (8 cases each).

Nodal involvement (N1 status according to AJCC, [3]) was observed in Osteosarcoma, Angiosarcoma, Follicular Dendritic Cell Sarcoma, Interdigitating Cell Sarcoma, Leiomyosarcoma, Malignant Fibrous Histiocytoma and Rhabdomyosarcoma.

Distant metastasis (M1 status according to AJCC, [3]) was found in two cases of Angiosarcoma and Osteosarcoma and in one case of Malignant Fibrous Histiocytoma and Rhabdomyosarcoma.

### 3.4. Management Strategy

The main modality of treatment for this tumor was surgical resection. It was performed in 77 cases (77/88, 87.5%). In particular, among patients that underwent surgery, total parotidectomy was the most common surgical procedure performed in 35 cases (35/77, 45.4%). The surgery was limited to the superficial lobe of the parotid gland in 24 patients (24/77, 31.2%), and radical parotidectomy with facial nerve sacrifice was performed in 18 patients (18/77, 23.4%). 

The surgery on cervical lymph nodes was performed in 27 patients (27/88, 30.7%). Twenty-two patients (22/27, 81.5%) underwent selective neck dissection and five (5/27, 18.5%) radical neck dissection. In 61 patients (61/88, 69.3%), neck dissection was not performed due to the absence of nodal involvement at the preoperative imaging.

Adjuvant therapies after surgical treatment were administered in 53 patients (53/77, 68.8%). These consisted of radiotherapy (RT) in 26 cases (26/53, 49.1%), a combination of RT and chemotherapy (CT) in 23 (23/53, 43.4%), and only CT in four cases (4/53, 7.5%). 

The neoadjuvant combination of RT and CT was performed only in four cases (4/77, 5.2%). 

In eleven cases (11/88, 12.5%), surgical treatment was not performed. Four cases (1 Angiosarcoma, 1 Rhabdomyosarcoma, 1 Osteosarcoma and 1 Malignant Fibrous Histiocytoma), were treated with CT alone due to distant metastasis. One case (Leiomyosarcoma T4N0M0) was treated with RT and CT due to comorbidity of the patient. Six children with Rhabdomyosarcoma (M0) were treated with either RT and CT in five cases or CT alone in one case.

The different therapeutic strategies are resumed in Table 4.

### 3.5. Survival Outcomes

In this review, we also analyzed the oncological outcome of parotid gland sarcomas. The overall survival (OS) of all enrolled cases was estimated by the Kaplan–Meier survival curve (Figure 2). The OS estimated considering all histological subtypes was 52% at 5 years and 34.1% at 10 years. 

The size/extension of the tumor (T according to AJCC, [3]) was the most important prognostic factor. Figure 3 shows Kaplan-Meier curves for each T stage. The OS for T1 stage tumor at 5 years of follow-up was 80.0%, while those for T2, T3 and T4 at 5 years were 66.5%, 56.7% and 33.3%, respectively. 

The histologic type of the sarcoma was another important prognostic factor. Indeed, Kaplan–Meier survival curves for the most frequent sarcomas (Figure 4) revealed that prognosis at 5 years was better for Liposarcoma (OS = 80.0%) and Synovial Sarcoma (OS = 78.8%) and worst for Angiosarcoma (OS = 23.4%). The OS for Leiomyosarcoma and Malignant Fibrous Histiocytoma at 5 years was 58.3% and 40%, respectively.

### 3.6. Multivariate Analysis

Multivariate analysis was performed including age, sex, tumor location in the parotid gland (superficial, deep or both), TNM according to AJCC [3], tumor histologic type, surgery performed on the tumor, surgery performed on lymph nodes, use of adjuvant therapy and use of neo-adjuvant therapy.

TFS as a dependent variable showed surgery (total or radical parotidectomy) performed on the tumor was the only statistically significant parameter (*p* = 0.0008). In particular, the analysis showed that the best modality of treatment was total parotidectomy and radical parotidectomy with facial nerve sacrifice in the locally advanced cases. Other variables had weak impact. Specifically, the age (younger) and the use of adjuvant therapy showed borderline significative values (*p* = 0.05 for both variables). Overall regression *p*-value was 0.008 and the power of the test was strong (0.9). Similarly, considering TSS as a dependent variable, the type of surgery was again the unique variable with strong statistical significance (*p* < 0.0001). Interestingly, borderline significance (*p* = 0.05) was observed for lymph node metastasis. Overall regression *p*-value was 0.00001, and the power of the test was strong (0.9).

## 4. Discussion

Sarcomas are a very heterogeneous group of tumors that may affect every organ in the body including the salivary glands in which they are extremely uncommon. Three reviews on this topic have been published since 1986. In the first, Auclair et al. [5] reported a total of 98 cases, 67 cases from the Armed Forces Institute of Pathology files and 31 previously reported cases in the literature. Based on their analysis, sarcomas of the salivary glands were found to behave identically to their soft tissue counterparts as prognosis was related to the size of the tumor and to the sarcoma’s histotype and grade. In 1991, Luna et al. [7] presented the MD Anderson Cancer Center experience during the period from 1945 to 1985 and performed a review of the English-language literature through January 1990. The most recent review was published in 2013 by Cockerill et al. [8]. The authors described 17 cases and reviewed the literature from January 1990 to July 2010, identifying 170 previously reported cases. Their analysis revealed that the parotid gland was largely the most commonly involved salivary gland (82.0%), Rhabdomyosarcoma was the most common histotype, that about three quarters of the patients were treated with some form of surgical resection and that about half of them received CT and/or RT.

Our review focused on the parotid gland. Rhabdomyosarcoma was confirmed to be the most common tumor. However, compared to the study of Cockerill et al. [8], in which it represented 13% of cases and showed a median age at diagnosis of 7 years, in our analysis, it represented 22% of cases and the median age was 18.3 years. As in the study of Cockerill et al. [8], the majority of patients were treated with some form of surgical resection and about half of them received CT and/or RT. Of note, our study is in agreement with the previously published reviews [6,7,8] also in terms of prognostic factors. Indeed, the size/extension of the tumor (T according to AJCC [3]) and the histologic type of sarcoma are the most important prognostic factors. The OS for T1, T2, T3 and T4 tumors at 5 years was 87.5%, 67.3%, 45.9% and 28.1%, respectively and, based on the histology, the OS at 5 years varied significantly for the different types of sarcomas. For example, at 5 years, it was 80.0% and 78.8% for liposarcoma and Synovial Sarcoma and 23.4% for Angiosarcoma.

Due to the rarity of the sarcomas in the parotid gland and also to their histological heterogeneity there is not a well-established consensus on their management [8]. Generally, surgery is the main treatment for primary parotid gland sarcomas. The surgical techniques are superficial, total or radical parotidectomy with facial nerve sacrifice. Our multivariate analysis demonstrates that total or radical parotidectomy is the best modality of treatment whatever the histological type and the size /extension of the tumor. Indeed, it is the only parameter that significantly affects the long-term survival. However, as properly noted by Cockerill and colleagues [8], “each histological subtype may behave differently, and treatment plans should be tailored accordingly”. For example, even though our study demonstrates that the role of CT is marginal in parotid gland sarcomas, it should be used for patients with histologic types that are known to be sensitive to CT such as Rhabdomyosarcoma, in which, in association to RT, it represents an effective treatment approach, and, as neoadjuvant treatment followed by surgical resection (and adjuvant RT), Osteosarcoma [8].

The role of RT in parotid gland sarcomas has never been analyzed in detail. Indeed, there are not systematic reviews or meta-analysis on the role of RT in parotid gland sarcomas. For epithelial malignant neoplasms of the parotid gland (mucoepidermoid carcinoma, carcinoma ex pleomorphic adenoma, adenoid cystic carcinoma, salivary duct carcinoma, and acinic cell carcinoma), adjuvant RT is taken into consideration in high histological grade, microscopic or macroscopic residual disease, perineural or lympho-vascular invasion, T3-4 stage disease and/or positive lymph nodes [15,16,17]. Typically, malignant mesenchymal tumors are described as not sensitive to RT because neoplastic cells, through several mechanisms, may escape from RT-mediated cell death [18]. However, as sarcomas of the salivary glands frequently recur, adjuvant RT may have some benefits, at least in the local control of the disease [8].

Lymph node metastasis from sarcomas is uncommon [19]. Indeed, in our study, we found regional lymph node involvement in only 7 out of 88 patients (7.9%). Despite this, about one third of the patients underwent (selective/radical) lymph node dissection. In addition, our analysis revealed a borderline statistical significance of lymph node metastasis (*p* = 0.05) on TSS. These findings may open questions about the opportunity to perform lymph node dissection when a parotid gland sarcoma is diagnosed, and the lymph nodes are free of disease at the preoperative staging. This suggests that alternative strategies [e.g., close follow-up by imaging (each 3–6 months for the first 3 years) and evaluation of sentinel lymph nodes] should be pursued.

In our study, age and use of adjuvant therapy also showed a borderline statistical significance (*p* = 0.05) on TFS. Consequently, especially in the young, it may be worth considering adjuvant treatment after (total or radical) parotidectomy.

As it emerges from our study and from the literature [6,7,8,20], surgery is the mainstay approach for localized disease and CT is for patients with metastatic disease. However, great advancements in the characterization of the molecular pathogenesis of sarcomas has led to the development of new therapeutic strategies whose results are promising at least in patients with certain sarcoma histotypes [20,21].

### Study Limitation

This study presents some limitations. First, the conclusions are based on the works of previous authors, and, because the methods used were different, this can induce a bias. Secondly, we did not register the protocol of our review to PROSPERO. Third, K-agreement was not calculated during the selection process. Finally, there was an unbalance in the histological diagnosis (e.g., Rhabdomyosarcoma was reported two times more than Synovial Sarcoma). This could have an impact on the results of the study, even though we considered the entire class of sarcomas. In spite of these limitations, we feel that our study may provide potentially useful information for physicians and surgeons involved in the treatment of sarcomas of the parotid gland and for patients diagnosed with them.

## 5. Conclusions

Primary sarcomas of the parotid gland are very rare malignancies. They can present at any age, from childhood to old age, have locally aggressive behavior and a low tendency to metastasize to the lymph nodes of the neck. Size/extension (T according to AJCC [3]) at the diagnosis and the sarcoma’s histotype are the most important prognostic factors. The multi-linear regression analysis showed that surgery (total or radical parotidectomy) performed on the primary tumor was the only parameter that significantly affected TFS and TSS. Consequently, when feasible based on the stage of the disease and regardless of the sarcoma’s histotype, total or radical parotidectomy should be performed at any age. Because lymph node metastases have rarely been reported in sarcomas, including those involving the parotid gland, alternative strategies (e.g., close follow-up by imaging and evaluation of sentinel lymph nodes) should be pursued before performing lymph node (selective/radical) dissection.

## Figures and Tables

**Figure 1 cancers-14-04862-f001:**
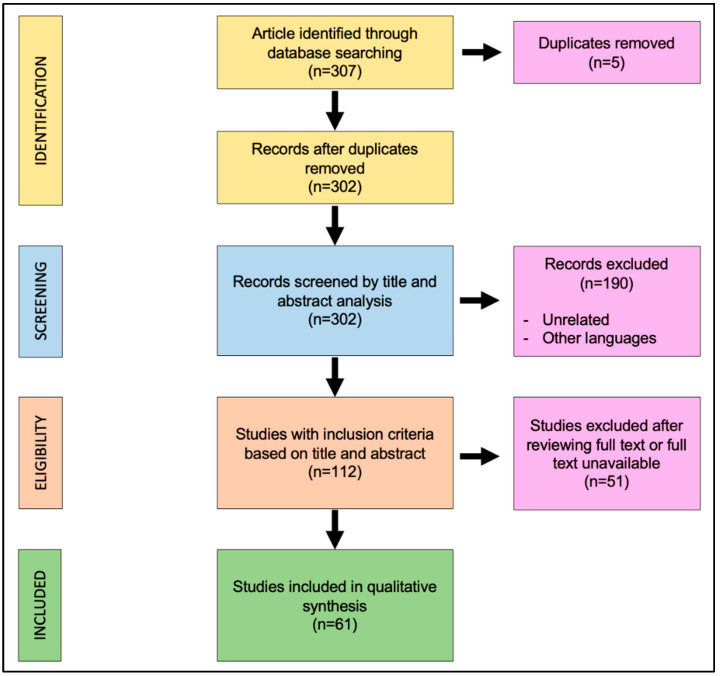
Preferred Reporting Items for Systematic Reviews and Meta-Analyses (PRISMA) diagram followed in this review. The diagram shows the information flow through the different phases of the review and illustrates the number of records that were identified and included.

**Figure 2 cancers-14-04862-f002:**
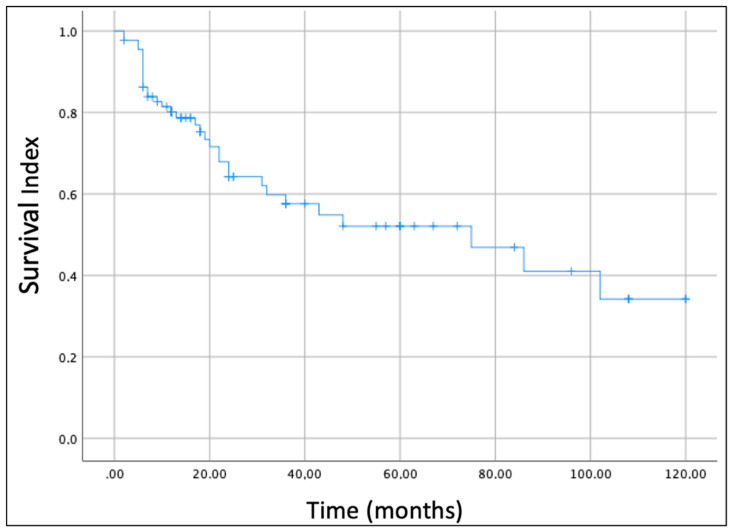
Kaplan-Meier curve and global overall survival (OS) of the patients with parotid gland sarcoma included in this study.

**Figure 3 cancers-14-04862-f003:**
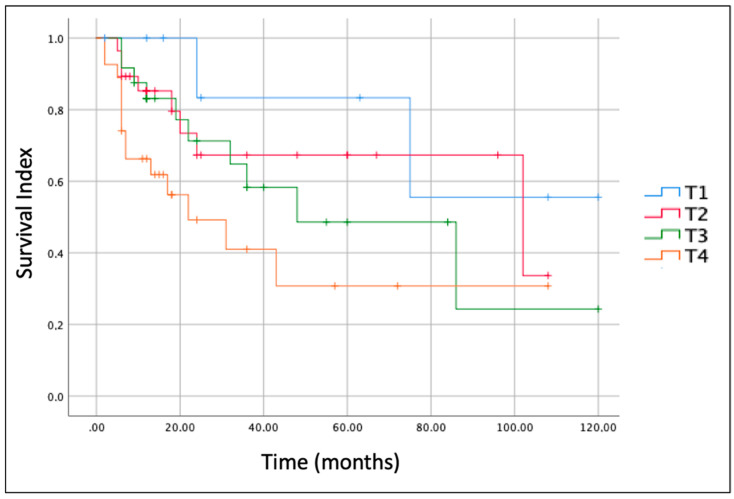
Kaplan-Meier curves and overall survival (OS) of the patients with parotid gland sarcoma included in this study according to the different size/extension (T according to AJCC, [3]) of the tumor.

**Figure 4 cancers-14-04862-f004:**
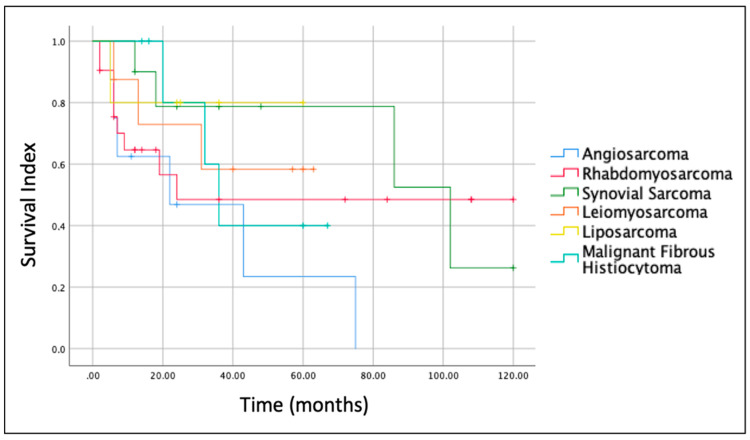
Kaplan-Meier curves and overall survival (OS) of the most frequent histological type of parotid gland sarcomas included in this study.

**Table 1 cancers-14-04862-t001:** Demographics features and clinical presentation of the parotid gland sarcomas included in this study.

Characteristics	N (% or Range)
Sex	
Male	48 (54.6%)
Female	40 (45.4%)
Total	88
Mean age (range)	39.4 (1–87)
Initial main symptoms	
Swelling	52 (59.1%)
Trismus	17 (19.3%)
Facial palsy	6 (6.8%)
Otalgia	5 (5.7%)
Skin fistula	5 (5.7%)
Ear fullness	3 (3.4%)

**Table 2 cancers-14-04862-t002:** Tumor location and TNM according to AJCC [3] of the parotid gland sarcomas included in this study.

Characteristics	N (%)
Tumor location	
Superficial lobe	49 (55.7%)
Deep lobe	7 (8.0%)
Superficial + deep lobe	21 (23.9%)
Parapharyngeal space	8 (9.1%)
Skull base	3 (3.4%)
TNM according to AJCC [3]	
T	
T1	9 (10.2%)
T2	28 (31.8%)
T3	24 (27.3%)
T4	27 (30.7%)
N	
N0	81 (92.0%)
N1	7 (8.0%)
M	
M0	82 (93.2%)
M1	6 (6.8%)

**Table 3 cancers-14-04862-t003:** Histological type of the parotid gland sarcomas included in this study.

Histological Type of Sarcoma	N (%)
Rhabdomyosarcoma	21 (23.9%)
Synovial Sarcoma	10 (11.4%)
Angiosarcoma	8 (9.1%)
Leiomyosarcoma	8 (9.1%)
Malignant Fibrous Histiocytoma	7 (8.0%)
Liposarcoma	5 (5.7%)
Adamantinoma-like Ewing Sarcoma	4 (4.6%)
Follicular Dendritic Cell Sarcoma	4 (4.6%)
Osteosarcoma	4 (4.6%)
Kaposi’s Sarcoma	3 (3.4%)
Low-grade Fibromyxoid Sarcoma	3 (3.4%)
Dermatofibrosarcoma Protuberans	2 (2.3%)
Ewing Sarcoma	2 (2.3%)
Extraskeletal Myxoid Chondrosarcoma	2 (2.3%)
Interdigitating Cell Sarcoma	2 (2.3%)
Chondrosarcoma	1 (1.1%)
Epithelioid Sarcoma	1 (1.1%)
Histiocytic Sarcoma	1 (1.1%)

**Table 4 cancers-14-04862-t004:** Main therapeutic approaches for the parotid gland sarcomas included in this study.

Therapeutic Approaches	N (%)
Surgery on “T”	77 (87.5% = 77/88)
Total parotidectomy	35 (45.4% = 35/77)
Superficial parotidectomy	24 (31.2% = 24/77)
Radical parotidectomy (with facial nerve resection)	18 (23.4% = 18/77)
None	11 (12.5% = 11/88)
Surgery on “N”	27 (30.7% = 27/88)
Selective neck dissection	22 (81.5% = 22/27)
Radical neck dissection	5 (18.5% = 5/27)
None	61 (69.3% = 61/88)
Adjuvant therapy	53 (68.8% = 53/77)
RT	26 (49.1% = 26/53)
CT	4 (7.5% = 4/53)
CT + RT	23 (43.4% = 23/53)
None	24 (31.2% = 24/77)
Neoadjuvant therapy	4 (5.2% = 4/77)
CT	4 (100% = 4/4)
None	73 (94.8% = 73/77)
Other therapy	11 (12.5% = 11/88)
CT	5 (45.5% = 5/11)
CT + RT	6 (55.5% = 6/11)

## Data Availability

Not applicable.

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
