# Peer review of "Systematic Review of Parotid Gland Sarcomas: Multi-Variate Analysis of Clinicopathologic Findings, Therapeutic Approaches and Oncological Outcomes That Affect Survival Rate"

_cancers, 2022, doi:10.3390/cancers14194862_

Round 1
Reviewer 1 Report
The paper addressed a topic of limited interest because of the paucity of sarcomas from the parotid gland;
The collection of literature data and its evaluation might represent an upgrade when compared with previous published paper. The paper is well written and collecting data from the few works available in the current english literature addressing this issue gives a global and complete informations about epidemiology, therapy and prognosis in patients affected by parotid gland sarcomas.
The paper needs to be improved in English language and I found two auto-references not appropriate
Author Response
REVIEWER #1
The paper addressed a topic of limited interest because of the paucity of sarcomas from the parotid gland;
The collection of literature data and its evaluation might represent an upgrade when compared with previous published paper.
The paper is well written and collecting data from the few works available in the current English literature addressing this issue gives a global and complete information about epidemiology, therapy and prognosis in patients affected by parotid gland sarcomas.
The paper needs to be improved in English language and I found two auto-references not appropriate
We thank the reviewer for the careful review and for the favorable comments. The manuscript has been reviewed by a native English speaker for grammar and language improvement. The two auto-references have been removed.

Reviewer 2 Report
1. Introduction is poor on content. Please, add more information. First, 3 reviews on the topic have been published. Highlight at the end of the introduction the limitation of these current studies and the novelty of this work. Add in the introduction also, more information about evolution and classification of such sarcomas based on current classification systems like WHO and their staging AJCC?
2. Search strategy has to be well indicated, reporting the search terms input in each database. Also, Web of Science should be also screened.
3. Indicate last day search
4. Some minor grammar errors should be corrected (Clinical series, case report, observational, longitudinal and clinical trials was/WERE considered for the systematic review)
5. K-agreement has to be calculated during selection process
6. In the methods OS is introduced but in the abstract is reported as tumor-free survival and tumor-specific survival. These has to be defined
7. PROSPERO registration is missing
8. Study quality should be changed to risk of bias assessment and proper rob tool has to be adopted based on each study design
9. Methods about multivariate analysis are not descripted
Author Response
REVIEWER #2
- Introduction is poor on content. Please, add more information. First, 3 reviews on the topic have been published. Highlight at the end of the introduction the limitation of these current studies and the novelty of this work. Add in the introduction also, more information about evolution and classification of such sarcomas based on current classification systems like WHO and their staging AJCC?
We thank the reviewer for these comments. The introduction has been expanded keeping them into account. Please see the revised version of the manuscript.
- Search strategy has to be well indicated, reporting the search terms input in each database. Also, Web of Science should be also screened.
We reviewed 4 platforms and we thought that Web of Science could not add additional articles for this paper; we will consider your suggestions for our future works.
- Indicate last day search.
We added this info in the material and method section.
- Some minor grammar errors should be corrected (Clinical series, case report, observational, longitudinal and clinical trials was/WERE considered for the systematic review).
We corrected the sentence accordingly with your suggestions.
- K-agreement has to be calculated during selection process
We thank the reviewer for the comment. Cohen-coefficient is generally used for original studies. We did not do it in this review and we added its lack as study limitation.
- In the methods OS is introduced but in the abstract is reported as tumor-free survival and tumor-specific survival. These has to be defined.
Sorry for the mistake. We now corrected the abstract changing the two terms with the appropriate OS.
- PROSPERO registration is missing.
Yes, this review has not be registered in PROSPERO. Unfortunately, it is not possible to register a protocol in PROSPERO if the search/analysis of the literature has already initiated. Indeed, “Submissions must be made before data extraction commences (from October 2019)” (https://www.crd.york.ac.uk/PROSPERO/#aboutpage). In our case, search/analysis were started after October 2019. Therefore, we added this absence as limitation of the study.
- Study quality should be changed to risk of bias assessment and proper rob tool has too be adopted based on each study design.
We thank the reviewer for the comment. To assess the quality of the studies included in the analysis, we used the standards from Wasserman et al. (ref. 14 of the revised version of the manuscript) and we added the results in the paper (range, median, SD).
- Methods about multivariate analysis are not descripted.
We added all necessary info about multivariate in the statistical method section.

Round 2
Reviewer 2 Report
This work is lacking of standard and widely adopted criteria for conducting systematic review. Authors only partially followed my suggestions. In the present form this paper, is not suitable for publication in this Journal